# Role of Polymeric Immunoglobulin Receptor in IgA and IgM Transcytosis

**DOI:** 10.3390/ijms22052284

**Published:** 2021-02-25

**Authors:** Hao Wei, Ji-Yang Wang

**Affiliations:** 1Department of Immunology, School of Basic Medical Sciences, Fudan University, Shanghai 200032, China; weihao@fudan.edu.cn; 2Department of Clinical Immunology, Children’s Hospital of Fudan University, Shanghai 201102, China; 3Department of Microbiology and Immunology, College of Basic Medical Sciences, Zhengzhou University, Zhengzhou 450001, Henan, China

**Keywords:** polymeric immunoglobulin receptor (pIgR), immunoglobulin transcytosis, immunoglobulin A (IgA), immunoglobulin M (IgM), joining chain (J chain), marginal zone B and B1 cell-specific protein (MZB1), antibody secretion, mucosal immunity

## Abstract

Transcytosis of polymeric IgA and IgM from the basolateral surface to the apical side of the epithelium and subsequent secretion into mucosal fluids are mediated by the polymeric immunoglobulin receptor (pIgR). Secreted IgA and IgM have vital roles in mucosal immunity in response to pathogenic infections. Binding and recognition of polymeric IgA and IgM by pIgR require the joining chain (J chain), a small protein essential in the formation and stabilization of polymeric Ig structures. Recent studies have identified marginal zone B and B1 cell-specific protein (MZB1) as a novel regulator of polymeric IgA and IgM formation. MZB1 might facilitate IgA and IgM transcytosis by promoting the binding of J chain to Ig. In this review, we discuss the roles of pIgR in transcytosis of IgA and IgM, the roles of J chain in the formation of polymeric IgA and IgM and recognition by pIgR, and focus particularly on recent progress in understanding the roles of MZB1, a molecular chaperone protein.

## 1. Introduction

The mucosa is an extensive layer of protection for the respiratory, gastrointestinal and urogenital tracts and other secretory glands such as the mammary glands. Separating the internal and external environments, the mucosa is constantly exposed to a wide variety of microorganisms and extrinsic molecules including bacteria, viruses, fungi and toxins. In human beings, the total surface area of the epithelial barrier is about 400 m^2^ [1]. Protection of the mucosal epithelium is provided by a vast network of proteins, molecules and cells, which are collectively termed as mucosal immunity [2].

Among a myriad of effectors in mucosal immunity, polymeric immunoglobulins IgA and IgM are of particular importance. Immunoglobulin (Ig) is the antigen-recognition molecule derived from B cells, while antibodies are secreted versions of immunoglobulin. An antibody is formed by two identical pairs of heavy and light chains joined together by disulfide bonds. There are five main classes of antibodies: IgA, IgD, IgE, IgG and IgM, which can be distinguished by their heavy chains. Only IgA and IgM can polymerize. The formation of IgA dimers and IgM pentamers is mediated by the joining chain (J chain), while IgM hexamers can be formed in the absence of J chain. Polymerized IgA, and to a lesser extent IgM, protect the mucosal surfaces from infection. In addition, small amounts of IgD are secreted into the mucosal surfaces of oral, nasopharyngeal and lachrymal areas [3]. Daily production of IgA in humans reaches 40 to 60 mg per kg of bodyweight, which is higher than that of all of the other immunoglobulin isotypes combined [4].

Delivery of antibodies to the mucosal surfaces and secretion in milk requires transport across epithelial layers. The polymeric immunoglobulin receptor (pIgR) recognizes the J chain region of polymerized IgA and IgM and transports the antibodies across the epithelial cell. Following proteolytic cleavage of pIgR, polymerized Ig is secreted and released into the luminal space. Since its function was first discovered in the 1980s, J chain, along with the molecular details of Ig polymerization, has largely been overlooked in the field of immunological research. It was only in the last decade when researchers started to take notice of marginal zone B and B1 cell-specific protein (MZB1) [5,6,7], a novel regulator of J chain-mediated Ig polymerization that precedes pIgR-mediated transcytosis.

In this review, we present an overview of how pIgR mediates transcytosis and the consequences of pIgR deficiency. We also expand on the molecular details of J chain binding to Ig polymers and recognition by pIgR based on results from recently published structural studies. We further highlight the proposed roles of MZB1 in the polymerization of IgA and IgM and briefly summarize the latest reports that have implicated MZB1 in human diseases.

## 2. Polymeric Immunoglobulin Receptor (pIgR)

### 2.1. Structure and Expression of pIgR

Human pIgR is a type I membrane protein with high glycosylation levels. It has a molecular weight of about 83 kDa [8]. Structural insights of pIgR were first obtained when the cDNA encoding the receptor was cloned and sequenced. As a typical type I transmembrane protein, the structure of pIgR can be categorized into three portions: an extracellular portion (620 amino acids, a.a.), a transmembrane portion (23 a.a.), and a cytoplasmic portion (103 a.a.) [9]. The extracellular portion of pIgR contains six domains [10]. Domains 1 to 5 are five tandem immunoglobulin-like domains that are involved in binding to IgA dimers or IgM pentamers. The cysteine residues in the extracellular portion, from which disulfide bonds may form, are conserved among human, mouse, rat and a few other mammalian species [11]. Domain 6, which is closest to the transmembrane portion, contains a highly conserved proteolytic cleavage site [12]. Compared to the extracellular portion, the intracellular portion of pIgR is more directly involved in intracellular sorting, endocytosis and transcytosis [13] (Figure 1). For structures of pIgR in vertebrates other than mammals, there have been several excellent articles covering the expression, structure and functions of pIgR in amphibians [14,15], fish [16,17,18], birds [19] and reptiles [20,21].

The human *pIgR* gene (NCBI gene ID: 5284) is located on the q32.1 region of chromosome 1. With a total of 11 exons, the human *pIgR* gene spans about 19 kb [22]. PIgR is expressed on epithelial cells of the gastrointestinal tract, respiratory tract and the skin, as well as on the glandular epithelial cells of the breast and liver [1,23,24,25]. A variety of immunological factors have been identified to upregulate expression of pIgR, including interleukin-1 (IL-1), interleukin-17 (IL-17), interferon-γ (IFN-γ), and tumor necrosis factor-α (TNF-α) [26,27,28,29]. Early studies also pointed out that interleukin-4 (IL-4), when acting in synergy with IFN-γ, can upregulate the expression of pIgR [30,31]. The effects of these cytokines on pIgR expression are mediated by transcription factors such as nuclear factor-κ light chain enhancer of activated B cells (NF-κB) and interferon regulatory factor-1 (IRF-1) [32,33], binding sites of which are located in the 5′-flanking region and intron 1 of *pIgR* gene [34].

From a functional perspective, upregulation of pIgR expression levels has been associated with bacterial, viral and chlamydial infections, where the immune system is activated and antibodies are produced and trafficked to fight off the pathogens [35,36,37,38,39]. Some pathogens have evolved strategies to utilize or suppress pIgR expression for the benefit of their infection. *Streptococcus pneumoniae*, *Candida albicans* and Epstein-Barr virus (EBV) can bind to pIgR, which aids their attachment to epithelial cells [40,41,42]. *Escherichia coli* and simian immunodeficiency virus (SIV) have been reported to downregulate pIgR expression, thereby evading the mucosal immune response [43,44,45]. The commensal microbiome also modulates pIgR expression. It was first reported when colonization of germ-free mice with a commensal bacterial strain *Bacteroides thetaiotaomicron* stimulated pIgR expression [46]. Later it was found that pIgR expression could be stimulated in vitro in HT-29 cells, a human intestinal epithelial cell line, when these cells were co-cultured with commensal bacterial strains from the family Enterobacteriaceae [33,47]. It was then proposed that microbial-associated molecular patterns (MAMPs) secreted from the commensal microbiome stimulate epithelial Toll-like receptors (TLRs), which triggers transcription of the *pIgR* gene by activating MyD88-dependent signaling pathways [23,25].

Interestingly, studies in mice have linked increased pIgR expression levels in submandibular glands to body exercise and heat acclimatization. Both studies have attributed this phenomenon to mild physiological stress, which might trigger an immune response [48,49]. In the past decade, modulation of *pIgR* expression, either elevated or reduced, has been increasingly reported in patients of cancer and metastasis, especially in hepatocellular and pancreatic cases [50,51,52,53,54,55,56]. It is possible that regulation of pIgR expression extends beyond the immune response. Further studies are required to elucidate the underlying mechanisms linking pIgR expression to cancer.

### 2.2. Functions of pIgR in Transcytosis of IgA and IgM

Epithelial cells form a layer of protection and insulation between the external and internal environments, defining the lumen of secretory organs and mucosal surfaces. These epithelial cells are polarized, which means the basolateral and apical membranes are of different compositions and functions. In order to move across the epithelial barrier, large solutes such as immunoglobulins must undergo transcytosis or the transcellular endosomal pathway [57]. The major function of pIgR, as its name suggests, is to bind polymeric immunoglobulins, thereby facilitating their transport across the epithelium [9].

Expressed on the basolateral surface of epithelial cells, pIgR, via its extracellular portion, binds to polymeric forms of IgA or IgM that are produced by local highly-differentiated plasma cells. Unlike other classes of antibodies (IgD, IgE, IgG), monomers of IgA and IgM can polymerize. Primarily, IgA forms dimers [58], whereas IgM forms pentamers and sometimes polymers of even higher orders [59]. This process of immunoglobulin polymerization is mediated by a special protein called joining chain (J chain) that binds to heavy chains of IgA and IgM through disulfide bonds at their *C*-terminal tailpieces [60]. Details of this process are discussed later in this article.

Transcytosis of IgA dimers and IgM pentamers is initiated once they bind to pIgR, and the Ig-pIgR complex is internalized into the cytoplasm of the epithelial cell via clathrin-mediated endocytosis, as was shown in an early in vitro study where pIgR was extrinsically expressed in MDCK cells [61]. Internalization of pIgR can occur even in the absence of its ligand [61]. The internalized Ig-pIgR complex travels along the endosomal pathway. The complex is first trafficked to the basolateral early endosome (EE), followed by transport to the common endosome (CE), before being sorted to the apical recycling endosome (ARE) that is localized beneath the apical epithelial membrane [62]. Sorting and targeting of pIgR throughout the endosomal transcytosis pathway is mediated by a signal of 17 membrane-proximal a.a. residues at the intracellular portion of the pIgR structure [63]. At the apical cell surface, the extracellular portion of pIgR, which binds to polymerized Ig molecules, undergoes endo-proteolytic cleavage at domain 6. The identity of the enzyme responsible for this cleavage remains obscure. The cleaved extracellular portion is referred to as the secretory component (SC). IgA dimers or IgM pentamers, which are originally bound by the SC, are released from the remaining transmembrane and intracellular portions of pIgR. The free, unbound SC-Ig polymer complexes are released as secretory Ig and diffuse into the mucus, where they act as an immunological barrier against infections by denying pathogens access to the epithelium [13,57,64]. This function of secretory Ig has been specifically termed as “immune exclusion” [65].

Apart from facilitating transport of the Ig polymers across the epithelium and their release into the mucus, SC, the cleaved extracellular portion of pIgR, has other critical functions. SC increases the stability of dimerized IgA, possibly by masking proteolytic cleavage sites within the IgA molecule, delaying the degradation of IgA by the host and bacterial enzymes in the mucus [66]. As SC is derived from pIgR, it is enriched in modifications such as *N*-glycosylation, so SC can help localize the secretory Ig complex in the mucus layer [67,68]. Even in the absence of Ig polymers, SC itself may bind and neutralize bacteria and toxins via its glycan moieties [1,69]. Cryo-EM structure of SC complexed with an IgA dimer showed that N65, N72, N168, N403, N451 and N481 are spatially away from any SC-IgA interaction surfaces, so glycosylation at these asparagine residues could be involved in the host and pathogen binding [70]. Functions of SC may be especially important for immunity in breast-fed infants, as the abundance of SC in both its free and Ig-bound forms has long been recorded in maternal milk [71,72,73].

### 2.3. Consequences of pIgR Deficiency

Studies on genetic knockout mice were pivotal in expanding our understanding of the functions of pIgR. The earliest studies of *pIgR^−/−^* mice were conducted in 1999. Epithelial transport of IgA was significantly reduced, although not ablated, in bile, feces and intestinal contents in *pIgR^−/−^* mice. Meanwhile, serum IgA levels were markedly increased in *pIgR^−/−^* mice [74,75]. These results demonstrated the essential roles of pIgR in transcytosis of IgA into the intestinal lumen, yet a small amount of IgA may be secreted via other pathways. Since the route for IgA transcytosis into the intestinal lumen is blocked in these genetically-deficient mice, IgA only has access to the blood. The increase in serum IgA levels might be further accounted for by increased numbers of plasma cells that secrete IgA, which was reported in both the lamina propria and the Peyer’s patch of *pIgR^−/−^* mice [76,77]. In addition, a lack of secretory IgA was observed in the pulmonary airways of *pIgR^−/−^* mice that developed signs similar to chronic obstructive pulmonary disease (COPD) [78,79,80]. These signs, which were caused by local infection and inflammation, were exacerbated by neutrophils of increased counts and activities [80].

Microbiota in the gut is altered as a result of pIgR deficiency. This was directly confirmed by comparing the intestinal microbiota between *pIgR^−/−^* and WT mice using 16s rRNA analysis [81]. Intestinal integrity was mildly compromised in *pIgR^−/−^* mice, which might be attributed to a slightly more severe bacterial insult in *pIgR^−/−^* mice [82,83]. This may explain the results from an early study, which showed that *pIgR^−/−^* mice were profoundly more sensitive to infection with *Salmonella typhimurium* via the fecal-oral route, and that bacteria excreted from *pIgR^−/−^* mice after *S. typhimurium* infection were more contagious for other mice [84], as the composition of the excreted bacterial population may differ between *pIgR^−/−^* and WT mice.

It is not surprising that pIgR deficiency has been extensively linked to inflammatory diseases in the gut, given the central role of secretory IgA in suppressing inflammation and maintaining homeostasis in the gut [85,86]. Dextran sulfate sodium (DSS)-induced colitis in mice is the most widely used animal model to study mechanisms of inflammatory bowel diseases (IBD) in humans, which mainly comprise ulcerative colitis and Crohn’s disease [87]. Morbidity and mortality of DSS-induced colitis were significantly enhanced in *pIgR^−/−^* mice [81]. Similarly, reduced levels of pIgR and secretory IgA in the gut, as a result of genetic deficiency in IL-17, were correlated with increased weight loss and more severe intestinal inflammation in mice following DSS administration [88]. Lower mRNA levels of *pIgR* in colonic mucosa have been proposed as a potential biomarker for the clinical diagnosis of IBD [89]. More recently, several cutting-edge studies compared whole-genome sequencing data from the colonic tissues of human IBD patients to those of healthy donors. Among these IBD patients, *pIgR* has been discovered as one of the most commonly shared sites of somatic mutations that are correlated with impaired protein functions, and these somatic mutations tend to accumulate with age [90,91,92]. Taken together, these data unequivocally illustrate the necessity of pIgR in transcytosis of polymeric immunoglobulins and thus in protection against inflammation.

## 3. Joining Chain (J Chain)

### 3.1. Structure and Expression of J Chain

J chain is a small polypeptide that mediates the polymerization of IgA and IgM. In the process of Ig transcytosis across the epithelium, J chain is essential in the recognition and binding of IgA and IgM polymers by pIgR [93]. In humans, there are two subtypes of IgA, IgA1 and IgA2, whereas in mice there is only one type of IgA, which more closely resembles to human IgA2 [94]. IgA1 and IgA2 differ mainly in their hinge region and the number of glycosylation sites. IgA1 is the predominant subtype in serum, while IgA1 and IgA2 are more evenly distributed in mucosal tissues, and relative proportions vary according to the specific site [95,96]. Human IgA1, IgA2 and mouse IgA all tend to form dimers when they bind to J chain and they are predominantly in the monomeric form when J chain is absent [97], although IgA1 and IgA2 oligomers up to pentamers have also been reported in the presence of J chain [98]. IgM forms pentamers when bound to J chain. Unlike IgA, IgM does not tend to form monomers in the absence of J chain. Instead, J chain-free conformations of IgM consist predominantly of hexamers and sometimes multimers of other orders without a consistent pattern, which has been described as “disordered oligomers” [60,99,100]. Only the J chain-bound forms of IgA and IgM are recognized by pIgR, undergo transcytosis, and get secreted into the mucus [97,101].

Structurally, J chain is an acidic polypeptide with a molecular mass of about 15 kDa and a length of 137 a.a. residues [102]. It includes eight cysteine residues, six of which form three intramolecular disulfide bonds (C13–C101, C72–C92, C109–C134), and the remaining two (C15 and C69) are involved in forming disulfide bonds with cysteine residues at the conserved 18-residue tailpieces of IgA and IgM heavy chains [103,104,105]. The *C*-terminal domain (25 a.a.) of J chain is involved in binding to the extracellular portion of pIgR. IgA remains bound to SC, which is cleaved off from pIgR, after Ig polymers are secreted into the mucus [106,107]. J chain is yet to be classified into a known family of proteins based on its primary structure or a.a. sequence, while a recent study has shown that the secondary structure of J chain comprises almost entirely of β sheets and loops [70]. From an evolutionary perspective, J chain is found in almost all species of jawed vertebrates with the exception of teleost. Its function in joining Ig monomers and binding to pIgR is conserved among jawed vertebrates [108].

The human *JCHAIN* gene (NCBI gene ID: 3512) is located on the q13.3 region of chromosome 4. With a total of 4 exons, the human *JCHAIN* gene spans about 9.8 kb [109,110]. Expression of J chain is recorded in all subtypes of plasma cells, not limited to those producing IgA or IgM [93]. At the transcription level, expression of J chain is activated by myocyte enhancer binding factor 2B (MEF-2B), a member of the B cell lymphoma 6 (BCL-6) protein complex [111], and inhibited by paired box protein 5 (Pax5) [112]. During plasma cell differentiation, B lymphocyte-induced maturation protein 1 (Blimp-1) downregulates Pax5, thereby relieving its inhibitory effect on J chain expression [113]. In plasma cells where J chain is not used for Ig polymerization, such as in IgG-secreting plasma cells, the unused J chain is degraded [114].

### 3.2. J Chain in IgA Polymerization, Binding to pIgR and Function

J chain is essential for the binding of IgA dimers to pIgR and their transcytosis across the epithelium. In *Jchain^−/−^* mice, a significantly lower ratio of IgA dimer to monomer was reported [115], and transepithelial transport of polymeric IgA was impaired [116]. Details of J chain binding to IgA, through covalent and non-covalent interactions, were resolved by cryo-EM [70]. J chain functions as a “clasp” between two Fcαs of IgA dimer via intermolecular disulfide bonds between C15 and C69 on J chain and C471 on either Fcα. Hydrophobic interactions are formed asymmetrically between β-hairpin structures in J chain and two Fcαs, at the top of one Fcα and the bottom of the other, respectively. This asymmetry is stabilized by interactions between β-sandwich structures in J chain and the Fcα tailpiece [70]. 

Binding of J chain to dimerized IgA facilitates its recognition by pIgR/SC. Lack of association between SC and IgA was reported in *Jchain^−/−^* mice [117]. In the absence of IgA dimer, pIgR adopts a “closed” conformation. A few hydrophilic residues in the “closed” pIgR make initial contact with J chain and the Fcα tailpiece. This triggers a large conformational change in pIgR, allowing it to contact extensively with J chain and both Fcαs. Domain 1 of pIgR interacts with the IgA dimer through non-covalent interactions, and domain 5 interacts with the IgA dimer through a single disulfide bond. C468 of pIgR, which originally forms an intramolecular disulfide bond with C502 of pIgR, forms an intermolecular disulfide bond with C311 on Fcα, locking pIgR into a bent, IgA-bound “open” conformation. Domains 2 to 4 of pIgR enhance binding affinity by providing correct spacing to allow the interactions of domain 1 and domain 5 with the IgA dimer, without making direct contact with the dimer. Consequently, pIgR/SC binds diagonally across the gap between two IgA monomers. Asymmetry in the IgA dimer, which is enforced by J chain, enables one-to-one binding of the kinked SC to the dimer [10,70]. 

During transcytosis, the extracellular domain (SC) of pIgR is proteolytically cleaved and remains bound to the IgA dimer, and they are secreted into the mucus together as the secretory IgA complex (sIgA) [13]. SIgA plays a pivotal role in mucosal immunity. The binding of sIgA to antigens, including bacteria, bacterial toxins, viruses and parasites, occurs either canonically via the complementarity-determining regions (CDRs) or non-canonically via glycan moieties [85,118]. The binding of sIgA to pathogens leads to agglutination and entrapment of pathogens in the mucus, preventing their adhesion to and subsequent penetration of the sub-mucosal epithelium. SIgA-agglutinated pathogens are excluded from mucosal surfaces, for example, by the peristaltic movement of the intestine [119]. SIgA also maintains intestinal homeostasis, potentially by promoting biofilm formation and gut colonization by commensal microbiota [120]. Retro-transcytosis of antigen-bound sIgA from the apical to the basal side of the epithelium, which is mediated by microfold (M) cells, enables uptake and delivery of antigens from the intestinal lumen to gut-associated lymphoid tissues [13,121]. Patients with IgA deficiency are more likely to show alterations in gut microbial taxa and develop disorders in the digestive tracts [122,123]. On the other hand, sIgA might be a cause for of autoimmunity. Anti-PDC-E2 IgA, transcytosis of which is mediated by pIgR, is recognized as a hallmark auto-antibody in primary biliary cirrhosis [124,125]. Similarly, pIgR expressed in salivary and lachrymal glands is a potential target of autoantibodies in Sjogren’s syndrome [126]. The presence of J chain-containing IgA was recently found in the dural sinuses in mouse meninges, where it protects the central nervous system by entrapping fungi cells [127]. These antibodies are clonally related to those in the gut [127], so the J chain-containing IgA produced in the gut lymphoid tissues may have functional implications beyond the mucosa.

### 3.3. J Chain in IgM Polymerization, Binding to pIgR and Function

IgM forms pentamers in the presence of J chain, and predominantly hexamers in the absence of J chain [60,99,100]. It was originally stated that IgM pentamers exhibited a stellate structure with 5-fold symmetry [101,128,129]. Nonetheless, recent studies have shown that a J chain-containing IgM pentamer from either mouse or human more closely resembles to a hexagon with a large gap in the middle, and that the J chain and the five Fcµ fragments are almost on the same plane [130,131,132]. Although the angle of the gap was measured to be 50 degrees by single-particle negative stain EM and 61 degrees by cryo-EM, a consensus has been reached that the gap is occupied by the J chain and it could be filled by a sixth IgM monomer when J chain is absent. Residues C14 and C68 on J chain form disulfide bonds with the C575 residues on adjacent Fcµ tailpieces [130,131]. Since C575 is also the site where the sixth IgM forms disulfide bridges with other IgM molecules in the IgM hexamer, J chain effectively thwarts the formation of IgM hexamer by occupying the site of covalent binding and supplanting the additional IgM monomer. A β-hairpin structure at the C terminal of J chain forms hydrophobic interactions extensively with Fcµ, further stabilizing the pentamer structure [131]. However, it needs to be pointed out that both IgM pentamers and hexamers were almost completely absent in *Jchain^−/−^* mice [133], so other factors might be involved in IgM secretion from B cells in vivo compared to cell culture systems.

J chain-containing IgM pentamer binds to pIgR for transcytosis. It was revealed that pIgR/SC docks perpendicular to the J chain-Fcµ plane. Similar to the scenario of IgA binding, IgM binding triggers a large conformation change in pIgR. Domains 1, 4 and 5 of pIgR rotate to accommodate the approaching IgM pentamer. A.a. residues in pIgR associate with residues in both the J chain and the IgM pentamer, forming hydrophobic interactions, salt bridges and hydrogen bonds. Given the indispensable roles of R105, A132 and Y134 in J chain for binding to SC, only J chain-containing IgM pentamers can bind to pIgR and thus be transported across the epithelium, whereas IgM hexamers cannot [131].

IgM forms the first line of defense against pathogen infections and, in some cases, auto-antigens as a part of the innate immune system in cooperation with other immune cells including mast cells, natural killer cells, dendritic cells and macrophages [59]. There are two distinct versions of IgM, namely the natural IgM and the adaptive IgM. The natural IgM is the predominant subtype, while the adaptive IgM accounts for 10–20% of all IgM antibodies in the serum in humans and mice. Natural IgM antibodies are reported to be produced by B-1 cells in the bone marrow and the spleen, and they are generally encoded by germline V gene segments with limited mutations [59]. By contrast, adaptive IgM antibodies are produced by post-germinal center plasma cells and memory B cells, so they have highly mutated V regions [59,134,135]. Although each natural IgM monomer has relatively low affinity to antigens as it normally does not undergo somatic hypermutation and affinity maturation, the presence of multivalent antigen-binding sites in IgM pentamers or hexamers confers high avidity, which is pivotal in the capacity of IgM for agglutinating pathogens by binding to specific antigenic motifs [101,136]. IgM exerts its functions primarily by fixing complement C1q, activating complement-dependent cytotoxicity and facilitating IgG-mediated opsonization [137]. It also has been proposed that IgM regulates immunity and tolerance through its specific Fc receptor (FcµR) [138].

Apart from canonical IgM functions in immune responses, J chain-containing IgM pentamers may serve as a carrier for apoptosis inhibitor of macrophage (AIM), which binds to the IgM pentamer solely in the presence of J chain [130]. It was proposed that AIM associates with the IgM pentamer by a 1:1 ratio, via a disulfide bond between C194 on AIM and C414 on Fc-Cµ3 of IgM. It was claimed that AIM is lodged at the edge of the topological gap in the IgM pentamer, which is not present in the IgM hexamer [130]. When bound to IgM, AIM remains inactive and it is protected from renal secretion [139]. When dissociated from IgM, AIM has been implicated in prevention and damage repair in diseases including multiple sclerosis, obesity, fatty liver disease, peritonitis, colorectal cancer, acute kidney injury and diabetic kidney disease [130,139,140,141,142,143,144,145,146]. Nevertheless, high levels of AIM have been reported mostly in the serum [139]. It remains obscure how the presence of AIM in IgM can be linked to transcytosis of its hypothetical carrier, the J chain-containing IgM pentamer, by pIgR across the epithelium into the mucus.

## 4. Marginal Zone B and B-1 Cell-Specific Protein (MZB1)

### 4.1. Early Studies on MZB1 Revealed Its Role in Assembly of IgM

The human *MZB1* gene (NCBI gene ID: 51237), also known as plasma cell-induced resident endoplasmic reticulum protein (pERp1), is located on the q31.2 region of chromosome 5, with a total of 4 exons. MZB1 of human and mouse origins show 71.4% a.a. identity. The sequences of MZB1 show only 21.6% overall conservation among mammals [147]. First identified in plasmacytoma cell lines as an 18 kDa ER-localized protein, *Mzb1* expression had been proposed to be specific for B and T lymphocytes in the spleen [5,6], yet microarray analysis of *Mzb1* expression in different subpopulations of immune and non-immune cells revealed that *Mzb1* is preferentially expressed in B cells and, to a lower extent, dendritic cells [148]. Expression of *Mzb1* in B cells was strongly upregulated during lipopolysaccharide (LPS)-induced B cell differentiation [5]. It can be directly activated by Blimp1, a master regulator of terminal B cell differentiation into plasma cells [149]. In dendritic cells, MZB1 may be critical for interferon α (IFN-α) secretion under stimulated conditions [150].

Co-immunoprecipitation (Co-IP) experiments showed that MZB1 binds to the heavy chain of IgM, via both disulfide bridges and non-covalent interactions [5,6]. When expression of *Mzb1* was silenced in the plasma cell line I.29μ^+^ by RNA interference, IgM secretion decreased and assembly between heavy and light chains of IgM was impeded [5]. MZB1 co-precipitated with the ER multichaperone complex glucose-regulated protein 94 (GRP94)-binding immunoglobulin protein (BiP) in the mouse plasmacytoma cell line Ag8.653 that does not express immunoglobulins [6], so MZB1 may assist IgM assembly as a molecular chaperone. Functions of classical chaperones BiP and GRP94 in protein folding in the ER have been reviewed in [151,152].

The role of MZB1 in IgM secretion was later confirmed in primary B cells, as downregulation of *Mzb1* led to impaired IgM secretion and overexpression of *Mzb1* potentiated IgM secretion in both marginal zone B cells (MZ B) and follicular B cells (FoB) in response to LPS [7]. MZB1 may also control Ca^2+^ homeostasis in B cells, possibly through interactions with the SERCA pump and ERp57, one of the regulators of SERCA, since knockdown of *Mzb1* enhanced depletion of Ca^2+^ from ER and extracellular Ca^2+^ influx in response to thapsigargin, and opposite effects on Ca^2+^ flow were observed upon overexpression of *Mzb1* [7]. In addition, MZB1 may have a role in integrin-mediated adhesion between MZ B cells and T cells [7]. Underlying mechanisms for these functions remain unclear.

Generation of MZB1-deficient mice enabled in vivo studies of MZB1 functions. Compared to wild type, *Mzb1^−/−^* mice showed a significant reduction in the amount of IgM secreted either from MZ B cells in response to T cell-independent antigens or from FoB cells in response to T-cell-dependent antigens [153]. Under conditions of ER stress, MZB1 associates directly with GRP94 and MZB1 is essential for GRP94 association with the µ heavy chain, as this interaction was almost ablated in *Mzb1^−/−^* splenic B cells [153]. Similarly, impaired IgM secretion in T cell-dependent responses was observed in another in vivo study with *Mzb1^−/−^* mice, yet secretion of IgG_1_ was not affected [154]. Taken together, results from these in vitro and in vivo studies indicated that MZB1 assists IgM secretion as a molecular co-chaperone of BiP and GRP94 and it enables proper folding of IgM heavy and light chains. It is worth further investigations whether MZB1 plays a part in IgM oligomerization, its association with J chain, and thus pIgR-mediated transcytosis.

### 4.2. Functions of MZB1 in IgA Polymerization

Apart from its role in IgM assembly, data from our study suggested that MZB1 may function as a molecular chaperone in dimerization of J chain-containing IgA [148], which is a prerequisite for the recognition of IgA by pIgR. MZB1 was required for efficient secretion of IgA, but not IgG, both in vivo and in vitro. Reduced levels of IgA, but not IgG, were detected in the serum of *Mzb1^−/−^* mice. Purified and cultured splenic B cells from *Mzb1^−/−^* mice secreted lower amounts of IgA, but not IgG, in response to stimulation with LPS, though class switching in these cells was not affected. Similar to the results obtained from splenic B cells, lower amounts of IgA, but not IgG, were secreted from MZB1-deficient Ag8.6532 (Ag8) cells after Ig genes were retrovirally transduced into the cell line. Secretion of IgA was restored to normal levels upon reexpression of MZB1 in MZB1-deficient Ag8 cells. In agreement with earlier studies, secretion of IgM was impeded under conditions of MZB1 deficiency in vivo and in vitro [148].

To mechanistically understand how MZB1 might regulate IgA secretion, a series of co-IP and immunoblotting experiments were conducted in Ag8 cells retrovirally transduced with Ig genes. First, MZB1 co-precipitated with IgA, but not IgG_1_. When the 18-residue secretory tailpiece of α heavy chain was deleted or the penultimate cysteine residue in the tailpiece was mutated, binding of MZB1 to IgA was ablated, so MZB1 potentially binds to IgA via its heavy chain tailpiece. Second, both the heavy chain and the light chain of IgA were more rapidly degraded in the absence of MZB1. In the absence of a light chain, the heavy chain was extremely stable, irrespective of MZB1 expression. It suggested that MZB1 stabilizes the assembled heavy chain-light chain IgA complex, rather than its components. By contrast, BiP interacts mainly with the heavy chain and it is not associated with the light chain. Finally, MZB1 was found to promote only the secretion of dimeric IgA, without any significant effects on that of monomeric IgA. Although binding of either MZB1 or J chain to IgA is dependent on the tailpiece, these two proteins did not co-precipitate [148]. Taking all these data into consideration, we proposed a model that BiP, MZB1 and J chain bind sequentially to α heavy chain. It allows efficient secretion of J chain-containing dimeric IgA, which then binds to pIgR, undergoes transcytosis across the epithelium, before being released into the mucus as sIgA (Figure 2).

Nonetheless, it is difficult to reconcile the data described above with structural studies on MZB1. First, the crystal structure of human MZB1 implied that all cysteines in the protein form buried intramolecular disulfide bonds. There is an apparent lack of exposed cysteines, from which intermolecular disulfide bonds with IgA or IgM may be formed [147]. It agrees with the data that MZB1 binds to IgA through non-covalent interactions [148], yet cysteine residues were deemed important in binding of MZB1 to IgM [5,6]. Second, BiP, but not MZB1, could prevent aggregation of thermally denatured citrate synthase, which is a classical in vitro assay to test chaperone activity, so MZB1 was not classified as a general molecular chaperone [147]. Third, MZB1 did not show a binding affinity for the *C*-terminal extension of IgA [147], which directly refutes results from our study [148]. It was thus proposed that MZB1 may not interact directly with IgA or IgM. There could be an unidentified protein that first binds to MZB1 in a cysteine-independent manner and subsequently interacts with IgA or IgM, which is dependent on disulfide bridges [147]. In our opinion, MZB1 chaperone activity might be specific, rather than general, for B cells, and the in vitro citrate synthase aggregation assay may not globally depict the biochemical environment within B cells. In addition, the *C*-terminal extension of IgA by itself in vitro might be topologically different from the *C*-terminal extension of IgA in the process of folding in vivo. It needs to be pointed out that structural details of MZB1 between human and mouse are likely to be critically different in terms of binding to Ig. A more convincing and definitive conclusion can only be reached after the crystal structure of mouse MZB1 is resolved and systemically compared with that of human.

MZB1 deficiency negatively affected the IgA response in vivo under acute but not homeostatic conditions. Completely normal IgA levels were detected in feces from *Mzb1^−/−^* mice at a steady state. Compared to wild type, *Mzb1^−/−^* mice showed significantly lower levels of fecal IgA only after intraperitoneal LPS injection or induction of acute colitis with oral DSS administration [148]. These data implied that a particular requirement of MZB1 exclusively occurs when IgA needs to be rapidly generated and secreted into the gut in order to suppress acute inflammation. Further studies are needed to clarify the mechanisms and whether it applies to other mucosal surfaces [155].

### 4.3. Implications of MZB1 in Human Diseases

MZB1 has been implicated in a wide range of human diseases, which might contribute to future development in diagnostics and therapeutics. Some diseases are correlated with the upregulation of *MZB1* expression, while reduced levels of *MZB1* have been reported in other cases. This is briefly summarized in Table 1.

Upregulation of *MZB1* has been reported in patients of several chronic auto-immune diseases, including periodontitis [156,157,158], systemic lupus erythematosus [159], multiple sclerosis, juvenile idiopathic arthritis-associated uveitis [160], Crohn’s disease [161], and, by extension, rejection of kidney transplant [162]. It is possible that *MZB1* overexpression leads to higher levels of local IgA and IgM secretion from plasma cells and their subsequent pIgR-dependent transcytosis in the mucus. Secreted IgA and IgM have the potential to become auto-antibodies that recognize self-antigens or the transplanted organ. As the expression of MZB1 is restricted mostly to B cells, MZB1 is a theoretically plausible target for therapies in autoimmunity and organ transport. A selective membrane-permeable inhibitor of MZB1, which is yet to be discovered, may increase the chance of improper folding of immunoglobulins, leading to its degradation and thus apoptosis of antibody-secreting plasma cells. It may cause fewer side effects than general immunosuppressants such as corticosteroids, sirolimus and cyclosporine.

It is difficult to find a pattern for the altered expression of *MZB1* in cancer. Overexpression of *MZB1* has been reported in patients of leukemia and lymphoma [163], pancreatic cancer [164], lung adenocarcinoma [165,166,167,168], multiple myeloma [169,170], estrogen receptor-positive breast cancer [171], and metastatic cutaneous melanoma [172]. Meanwhile, some suggested that MZB1 may be a tumor suppressor in hepatocellular carcinoma [173], gastric cancer [174], and colorectal adenocarcinoma [175]. Except for cases of leukemia, lymphoma and myeloma where elevated levels of *MZB1* may be linked to abnormalities in B cell proliferation, survival and differentiation, altered expression of *MZB1* is likely to be a byproduct of passenger mutations in most types of cancer. Anti-MZB1 treatment may not address the etiology or pathogenesis, yet it could be useful to measure the expression of *MZB1* for prognosis and diagnosis of cancer.

## 5. Conclusions

The mucosa is a large surface susceptible to pathogenic infections. Protection of mucosal surfaces is conferred by the mucosal immune response, which centers around polymeric antibodies IgA and IgM. Transport and secretion of polymeric IgA and IgM across the mucosal epithelium into the mucus is mediated by pIgR. PIgR binds to polymerized IgA and IgM at the basolateral membrane of the epithelial cells, which triggers endocytosis and transcytosis via the endosomal sorting pathway. At the apical membrane, pIgR is cleaved and SC complexes with IgA and IgM polymers to be secreted together as the functional secretory Ig. J chain is essential in the recognition and binding of Ig polymers by pIgR. Crystal and cryo-EM structures of J chain-containing secretory Ig provided insights into how J chain binds to Ig polymers and how it is directly involved in Ig binding with pIgR. The importance of pIgR in transcytosis of Ig polymers has been illustrated convincingly by data from knockout mice studies and human patients with genetic deficiencies.

MZB1, an ER-localized protein, facilitates the formation of IgA and IgM polymers. It was first identified as a co-chaperone of the BiP-GRP94 complex that is involved in the assembly of IgM. Data from a more recent study suggested that MZB1 promotes J chain binding to IgA, also in the capacity as a molecular chaperone, and that MZB1 is particularly important for transcytosis and secretion of IgA dimers into the gut under acute inflammatory conditions. However, MZB1 deficiency does not seem to affect IgA secretion under steady state in vivo. Data from a crystal structure of MZB1 do not match its proposed roles with respect to Ig polymerization. Despite these controversies, MZB1 has been implicated in various human diseases including auto-immune disorders and cancer, which are possibly linked to abnormal levels of antibody production, transcytosis and secretion. Further studies into the structure and functions of MZB1 could potentially provide guidance in developing new tools for diagnosis, prognosis and treatment of these diseases.

## Figures and Tables

**Figure 1 ijms-22-02284-f001:**
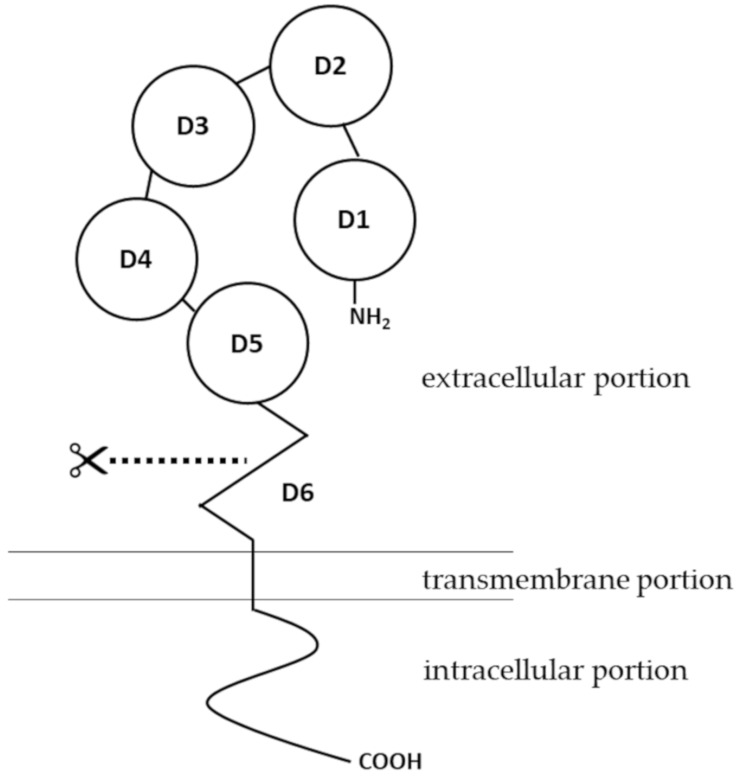
Structure of polymeric immunoglobulin receptor (pIgR). The pIgR contains an intracellular portion, a transmembrane portion and an extracellular portion. The extracellular portion has six domains. Extracellular domains 1 to 5 (D1–D5) are five tandem immunoglobulin-like domains that are involved in binding to Ig polymers. The extracellular domain 6 (D6) contains a site for proteolytic cleavage.

**Figure 2 ijms-22-02284-f002:**
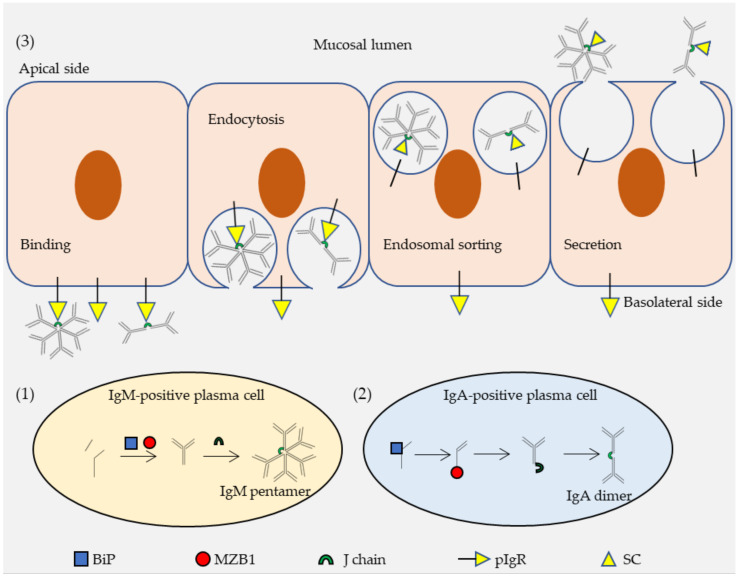
A proposed scheme for transcytosis of polymerized IgA (immunoglobulin A) and IgM (immunoglobulin M). (**1**) Formation of the IgM pentamer is mediated by marginal zone B and B-1 cell specific protein (MZB1) and J chain. MZB1 assists the assembly of IgM as a co-chaperone of the glucose-regulated protein 94 (GRP94)-binding immunoglobulin protein (BiP) complex. J chain aids formation of the IgM pentamer. J chain-containing IgM pentamers are secreted from IgM-positive plasma cells. (**2**) Formation of the IgA dimer is mediated sequentially by BiP, MZB1 and J chain. BiP binds and stabilizes the α heavy chain. MZB1 binds to the α heavy chain and stabilizes the light chain-heavy chain complex. J chain joins two IgA monomers. J chain-containing IgA dimers are secreted from IgA-positive plasma cells. (**3**) Polymeric immunoglobulin receptor (PIgR) mediates transcytosis of IgA dimers and IgM pentamers. IgA dimers and IgM pentamers, via the J chain, bind to the pIgR on the basolateral membrane of the epithelial cells. The Ig-pIgR complex undergoes clathrin-mediated endocytosis and is conveyed through the endosomal sorting pathway. PIgR is cleaved and its extracellular portion, the secretory component (SC), remains bound to the immunoglobulin (Ig) polymers. The SC-Ig polymer complex is released from the apical membrane of epithelial cells and secreted into the mucus. Note that the molecules and cells in this figure are not to scale.

**Table 1 ijms-22-02284-t001:** Implications of marginal zone B and B-1 cell specific protein (MZB1) in human diseases.

State of MZB1 Expression	Implicated Human Disease(s)	Reference
Elevated	Periodontitis	[156,157,158]
Elevated	Systemic lupus erythematosus, Rheumatoid arthritis	[159]
Elevated	Juvenile idiopathic arthritis-associated uveitis	[160]
Elevated	Crohn’s disease	[161]
Elevated	Rejection of kidney transplant	[162]
Elevated	Chronic lymphocytic leukemia, Follicular lymphoma, diffuse large B-cell lymphoma	[163]
Elevated	Borderline resectable pancreatic cancer	[164]
Elevated	Lung adenocarcinoma	[165,166,167,168]
Elevated	Multiple myeloma	[169,170]
Elevated	Estrogen receptor-positive breast cancer	[171]
Elevated	Cutaneous metastatic melanoma	[172]
Reduced	Hepatocellular carcinoma	[173]
Reduced	Malignant gastric cancer	[174]
Reduced	Colorectal adenocarcinoma	[175]
Elevated	Lung and skin fibrosis	[176]
Reduced	Multiple sclerosis	[177]

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
