# Peer review of "Role of Polymeric Immunoglobulin Receptor in IgA and IgM Transcytosis"

_ijms, 2021, doi:10.3390/ijms22052284_

Round 1
Reviewer 1 Report
The authors present a timely review of a topic that is of great importance to the field of mucosal immunology. The role of the polymeric Ig receptor (pIgR) has been underappreciated, and the current reviews fills this gap. It is well written and provides the audience with important updates. However, the following concerns should be addressed.
Major Concerns:
- The authors should include a figure depicting the structure of pIgR – especially since this protein is the main attraction of this review. This should be given as Figure 1.
- Table 1, entitled: “Implications of MZB1 in human diseases”, should be reorganized. The first column, “State of MZB1 expression”, should be sorted by “Reduced” or “Elevated” – instead of the current apparently random listing of either reduced or elevated expression of this molecule. The four conditions that seem to be linked to reduced MZB1 expression will be easier to be discerned by the readers.
- The authors did not discuss that IgM has two distinct versions, namely natural IgM and adaptive IgM. Only the former is produced by the B1 subset of B cells in mice and natural IgMs generally harbor germ line versions of the V gene segments with limited somatic hypermutations. This is not the case for adaptive IgMs. The authors should include this important aspect of IgM biology in their review.
- The authors should introduce the term “immune exclusion” to describe the trapping of incoming pathogens by SIgA and SIgM present in mucosal fluids. While the authors discuss this trapping function in several places, the specific term “immune exclusion” has not been used, as e.g., in lines 128-130, and in in the top paragraph on page 6.
Minor Concerns:
- Abstract, line 2 should read “…the epithelium and subsequent secretion into mucosal fluids are mediated by…” [not “subsequent secretion into the mucosa”]
- Introduction, line 1 should read “The mucosa is an extensive layer of protection for the respiratory, gastrointestinal, and urogenital tracts…” [delete “the epithelium” in line 23]
- Introduction, line 36: authors state that only IgA and IgM can polymerize, which is mediated by the joining chain (J chain). This is not the case for hexameric IgM, a polymeric form of IgM that arises in the absence of J chain expression. The statement in lines 35-36 should be rephrased to account for these facts regarding hexameric IgM.
- Page 2, line 46: should read “It was only in the last decade…” [not “It was only until the last decade…”]
- Page 2, line 47: at the end of the line, after (MZB1), the references for this protein should be added here.
- Page 2, line 61: insert an article to read “the structure of pIgR can be categorized…”
- Page 2, lines 79-83: the use of hyphenation is inconsistent; hyphens should be used for interleukin-1, interferon-γ, tumor necrosis factor-α, interleukin-4.
- Line 79 insert “and” in front of tumor necrosis factor-α.
- Lines 156-157: replace “symptoms” with “signs”. Symptoms in clinical medicine are subjective sensations, such as headache and nausea, which cannot be communicated by animals. In contrast, “signs” are disease manifestations that can be noticed by an observer.
- Line 362: replace “was” with “were” [use the plural].
- Insert the article “the” on line 390 to read “Firstly, the crystal structure of human MZB1…”
- Line 398: delete “with” to read “…, which directly refutes results from our study”.
- Legend to Figure 1, Line 430: “…in this figure are not to scale” [not “in scale”].
- Line 438: insert comma after “sclerosis, juvenile idiopathic arthritis–associated uveitis…”
- Line 470: insert “J-chain-containing secretory IgG” [insert hyphen between “chain” and “containing”].
- Line 475: insert comma after “MZB1” to read “MZB1, an ER-localized protein, …”
Author Response
Major Concerns:
- The authors should include a figure depicting the structure of pIgR – especially since this protein is the main attraction of this review. This should be given as Figure 1.
Figure 1 is now added in line 113, with legends in lines 114-116. It is mentioned in the text in line 73. The original figure 1 is now re-labelled as Figure 2, and relevant changes have been made in lines 428 and 461.
- Table 1, entitled: “Implications of MZB1 in human diseases”, should be reorganized. The first column, “State of MZB1 expression”, should be sorted by “Reduced” or “Elevated” – instead of the current apparently random listing of either reduced or elevated expression of this molecule. The four conditions that seem to be linked to reduced MZB1 expression will be easier to be discerned by the readers.
We have reorganized Table 1 as suggested (line 501).
- The authors did not discuss that IgM has two distinct versions, namely natural IgM and adaptive IgM. Only the former is produced by the B1 subset of B cells in mice and natural IgMs generally harbor germ line versions of the V gene segments with limited somatic hypermutations. This is not the case for adaptive IgMs. The authors should include this important aspect of IgM biology in their review.
This aspect of IgM biology is now included in lines 326-333.
- The authors should introduce the term “immune exclusion” to describe the trapping of incoming pathogens by SIgA and SIgM present in mucosal fluids. While the authors discuss this trapping function in several places, the specific term “immune exclusion” has not been used, as e.g., in lines 128-130, and in in the top paragraph on page 6.
This specific term “immune exclusion” is now included in lines 151-152. In line 281-282 (originally top paragraph on page 6), we use the verb “exclude”.
Minor Concerns:
- Abstract, line 2 should read “…the epithelium and subsequent secretion into mucosal fluids are mediated by…” [not “subsequent secretion into the mucosa”]
We have changed it accordingly in line 8.
- Introduction, line 1 should read “The mucosa is an extensive layer of protection for the respiratory, gastrointestinal, and urogenital tracts…” [delete “the epithelium” in line 23]
We have changed it accordingly in line 23.
- Introduction, line 36: authors state that only IgA and IgM can polymerize, which is mediated by the joining chain (J chain). This is not the case for hexameric IgM, a polymeric form of IgM that arises in the absence of J chain expression. The statement in lines 35-36 should be rephrased to account for these facts regarding hexameric IgM.
We have changed it accordingly in lines 36-37.
- Page 2, line 46: should read “It was only in the last decade…” [not “It was only until the last decade…”]
We have changed it accordingly in line 49.
- Page 2, line 47: at the end of the line, after (MZB1), the references for this protein should be added here.
We have changed it accordingly in line 50.
- Page 2, line 61: insert an article to read “the structure of pIgR can be categorized…”
We have changed it accordingly in line 64.
- Page 2, lines 79-83: the use of hyphenation is inconsistent; hyphens should be used for interleukin-1, interferon-γ, tumor necrosis factor-α, interleukin-4.
We have changed it accordingly in lines 81-83.
- Line 79 insert “and” in front of tumor necrosis factor-α.
We have changed it accordingly in line 82.
- Lines 156-157: replace “symptoms” with “signs”. Symptoms in clinical medicine are subjective sensations, such as headache and nausea, which cannot be communicated by animals. In contrast, “signs” are disease manifestations that can be noticed by an observer.
We have changed it accordingly in lines 178 and 179.
- Line 362: replace “was” with “were” [use the plural].
We have changed it accordingly in line 403.
- Insert the article “the” on line 390 to read “Firstly, the crystal structure of human MZB1…”
We have changed it accordingly in line 431.
- Line 398: delete “with” to read “…, which directly refutes results from our study”.
We have changed it accordingly in line 439.
- Legend to Figure 1, Line 430: “…in this figure are not to scale” [not “in scale”].
We have changed it accordingly in line 471, now Figure 2.
- Line 438: insert comma after “sclerosis, juvenile idiopathic arthritis–associated uveitis…”
We have changed it accordingly in line 479.
- Line 470: insert “J-chain-containing secretory IgG” [insert hyphen between “chain” and “containing”].
We have changed it accordingly in line 511.
- Line 475: insert comma after “MZB1” to read “MZB1, an ER-localized protein, …”
We have changed it accordingly in line 516.
Reviewer 2 Report
This review is very interesting and well written.
Some points are not discussed and are important:
J chain KO effects are not described.
IgD are also secreted, mainly in the oral area.
After secretion, a retrotranscytosis of IgA can occur.
Author Response
This review is very interesting and well written.
Some points are not discussed and are important:
1)J chain KO effects are not described.
J chain KO effects are now added in lines 251-252, 260-261, 312-315.
2)IgD are also secreted, mainly in the oral area.
Secretion of IgD is now added in lines 38-40.
3)After secretion, a retrotranscytosis of IgA can occur.
Retro-transcytosis of IgA is now added in lines 284-287.
Reviewer 3 Report
The authors have nicely reviewed the role of polymeric Ig receptor in the transcytosis of polymeric IgA and IgM from the basolateral surface to the apical side of the epithelium and subsequent secretion into the mucosa. They also focus on recent progress in understanding the roles of MZB1, a molecular chaperone protein.
This is a very well written review and very useful one for those interested in the role of polymeric Ig receptor in the transcytosis of polymeric IgA and IgM.
Some suggestions.
The authors have addressed changes in microbiota in the gut as a result of pIgR deficiency. Therefore, it would be of use to add information regarding the role of commensal microbiome in the regulation of pIgR. Perhaps, it would be useful to have a sub-section for this.
It would be also useful to discuss how pathogens subvert the pIgR system for their advantage.
It would also be useful to mention the role of pIgR in the transcytosis of IgA autoantibodies in autoimmune diseases (e.g anti-PDC-E2 IgA transcytosis by pIgR in primary biliary cirrhosis). It is of interest that pIgR is expressed in salivary and lacrimal glands, a target of autoantibodies in Sjogren’s syndrome.
Line 16: focus particularly on recent progresses
This should be-
focus particularly on recent progresses
Line 37: Daily production of IgA in human
This should be-
Daily production of IgA in humans
Not sure whether BiP and GRP94 has been expanded in the manuscript
Author Response
Some suggestions.
1)The authors have addressed changes in microbiota in the gut as a result of pIgR deficiency. Therefore, it would be of use to add information regarding the role of commensal microbiome in the regulation of pIgR. Perhaps, it would be useful to have a sub-section for this.
The role of commensal microbiome in the regulation of pIgR is added in lines 96-104.
2)It would be also useful to discuss how pathogens subvert the pIgR system for their advantage.
A brief discussion of how pathogens subvert the pIgR system for their advantage is added in lines 92-96.
3)It would also be useful to mention the role of pIgR in the transcytosis of IgA autoantibodies in autoimmune diseases (e.g anti-PDC-E2 IgA transcytosis by pIgR in primary biliary cirrhosis). It is of interest that pIgR is expressed in salivary and lacrimal glands, a target of autoantibodies in Sjogren’s syndrome.
We have added these in lines 289-292.
4)Line 16: focus particularly on recent progresses. This should be- focus particularly on recent progress
We have changed it accordingly in line 16.
5)Line 37: Daily production of IgA in human. This should be- Daily production of IgA in humans
We have changed it accordingly in line 40.
6)Not sure whether BiP and GRP94 has been expanded in the manuscript
We did not expand on BiP and GRP94, because we did not want to distract readers of this article with details of general protein folding. For readers who might be interested in this topic, we added a couple references in line 375-376.